# Voluntary Counseling and Testing, Antiretroviral Therapy Access, and HIV-Related Stigma: Global Progress and Challenges

**DOI:** 10.3390/ijerph19116597

**Published:** 2022-05-28

**Authors:** Elizabeth Afibah Armstrong-Mensah, Ato Kwamena Tetteh, Emmanuel Ofori, Osasogie Ekhosuehi

**Affiliations:** 1Department of Health Policy & Behavioral Sciences, School of Public Health, Georgia State University, Atlanta, GA 30303, USA; oekhosuehi1@student.gsu.edu; 2Laboratory Department, Metropolitan Hospital, Cape Coast P.O. Box 174, Ghana; atetteh3@student.gsu.edu; 3Department of Family Medicine, Komfo Anokye Teaching Hospital, Kumasi P.O. Box 1934, Ghana; emmo600gh@yahoo.com

**Keywords:** HIV prevention, voluntary counselling and testing, antiretroviral therapy, HIV-related stigma

## Abstract

To date, about 37 million people are living with the human immunodeficiency virus (HIV) and an estimated 680,000 people have died from acquired immune deficiency syndrome (AIDS) related illnesses globally. While all countries have been impacted by HIV, some have been significantly more impacted than others, particularly countries in sub-Saharan Africa. The purpose of this paper was to identify progress made in HIV prevention globally, particularly in the areas of voluntary counseling and testing (VCT) uptake, access to antiretroviral therapy (ART), and HIV-related stigma. With the development of ART, a cocktail of medications for the treatment of HIV, VCT uptake increased, as it became apparent that the medication would only be prescribed after an HIV diagnosis through testing. Widely considered a critical gateway to HIV prevention and treatment, VCT is being implemented in many countries, and as a result, about 38 million people living with HIV in 2018 had access to ART. Regardless of this success, major challenges still remain. We did an electronic search of 135 articles in English related to global HIV progress and challenges indexed in PubMed, ResearchGate, Google, and other search engines from 1998 to 2021. Sixty articles met the inclusion criteria for this paper. Data on trends in ART coverage were obtained from the Joint United Nations Programme on HIV/AIDS (UNAIDS) website. These data were used to show ART coverage globally in World Health Organization (WHO) regions. It was found that while global successes have been chalked in the areas of VCT uptake and ART coverage, HIV-related stigma has impeded greater success. This paper summarizes and discusses global successes and challenges in HIV prevention efforts in the past four decades with a focus on VCT, ART, and HIV-related stigma.

## 1. Introduction

Many more people are living with the human immunodeficiency virus (HIV) and the acquired immune deficiency syndrome (AIDS) today than when the incidence rate was at its peak in the 1990s and 2000s. This is due to several factors including the implementation of several public health interventions and campaigns, advances in science and technology such as HIV testing, and the development of antiretroviral therapy (ART). This notwithstanding, HIV continues to be a global public health issue [1]. Thus, as of 2020, there were 37.6 million people living with HIV (PLWH), 1.5 million new HIV infections, and 690,000 AIDS-related deaths globally [2]. That same year, women and girls accounted for about 50% of all the new HIV infections [2], and sub-Saharan Africa was home to 61% of all new infections [1]. It was in the attempt to address these issues and more that the global community launched the Millennium Development Goals (MDGs) in 2000 [3], and the Sustainable Development Goals (SDGs) in 2015 [4], with their respective targets of reversing HIV by 2015 (MDG 6) [3] and eliminating HIV by 2030 (SDG 3) [4].

After 40 years, some successes have been chalked, however, major challenges still remain. While voluntary HIV counselling and testing (VCT) uptake has increased globally, the lack of resources, the low national prioritization, low levels of awareness and access to VCT services, and overburdened health care providers still remain a challenge in certain parts of the world. Global ART coverage is uneven. While coverage in the world has significantly improved, gaps still remain in sub-Saharan Africa. In addition to the lack of medication to cure HIV, the stigma associated with the disease has made its prevention and potential eradication a global challenge [5]. Without addressing HIV-related stigma, SDG 3 will be a very distant reality, as HIV-related stigma has been identified as a major drawback in VCT uptake and ART utilization and adherence. 

We conducted an ecological study to identify progress made in HIV prevention after 40 years and the challenges that still remain. A comprehensive search of databases including PubMed, ResearchGate, and Google Scholar was done using keywords such as VCT, ART, and HIV-related Stigma. Of the reviewed 135 English articles published between 1992 and 2021, 60 met our paper’s inclusion criteria. The search was conducted between 14 October 2021, and 30 November 2021. This review summarizes and discusses progress made in HIV prevention globally after 40 years and the challenges that still remain, particularly in the areas of VCT uptake, access to ART, and HIV-related stigma. 

## 2. Global HIV Prevention Efforts and Progress Made after Four Decades

### 2.1. Voluntary Counselling and Testing

In 1994, the World Health Organization (WHO) and the Joint United Nations Program on HIV and AIDS (UNAIDS) proposed VCT as a strategy for the prevention and control of HIV [6,7]. VCT is a process that allows individuals to learn about their HIV status through pre-and post-test counseling and an HIV test [8]. Its objective is to get people to know their HIV status, reduce anxiety over possible infection, encourage safer sexual practices to prevent HIV transmission, facilitate safe disclosure of infection status and future planning, and to improve access to HIV prevention and treatment services [9]. Originally implemented as an individual-level clinic-based procedure, VCT is now also implemented as a community and couples-based approach with the aim of increasing uptake [10,11,12,13]. As its name implies, the clinic-based approach to VCT involves testing and counseling in clinical settings, while the community-based approach offers HIV testing and counseling at educational facilities and workplaces, during testing campaigns through mobile VCT services, at targeted events, and in people’s homes [11]. The couples-based approach allows two or more people who are in or planning to be in a sexual relationship to receive HIV testing and results together. This approach facilitates communication and the disclosure of HIV status [14,15,16,17].

In the beginning, global VCT uptake was very low as people questioned its benefits and the psychological effects on those who got tested [10]. However, following the development of ART, a cocktail of medications for the treatment of HIV, perceptions began to change when it became apparent that the medication would only be prescribed after an HIV diagnosis through testing [7,10]. Widely considered a critical gateway to HIV prevention and treatment, VCT is being implemented in many countries. In 2004, China introduced a free VCT program, [18] and by 2016, 588,970 people had received VCT, 580,974 had undergone HIV-antibody testing, and 12,340 were identified as infected with HIV [19]. Between 2004 and 2008, some sub-Saharan countries including Botswana and Lesotho instituted routine VCT backed by national ART programs [20]. In 2013, targeting the 18 countries most affected by HIV (Cameroon, Cambodia, China, the Democratic Republic of Congo, Egypt, Haiti, Honduras, Guatemala, India, Indonesia, Kenya, Mozambique, Nigeria, Russia Federation, South Africa, and Tanzania), the International Trade Unions Confederation (ITUC), International Organization of Employers (IOE), UNAIDS and the International Labor Organization (ILO) launched the VCT@WORK Initiative at the International Labor Conference. As a result of the launch, 4.1 million workers and their families got tested for HIV by 2016. Approximately 104,926 of this population tested positive for HIV, and about 103,286 were referred for treatment [21].

Advances in HIV testing technology in the last decade have further improved VCT. According to the WHO, “the earliest versions of HIV tests, conventional blood tests such as ELISA and Western Blot developed in the 1980s, were limited in scope and impact. The material and personnel costs of running conventional blood tests were difficult to implement in resource-poor settings” [22] (p. 116). The advent of and large-scale production of rapid HIV diagnostic tests, coupled with the fact that such tests require less-invasive oral fluid or finger-prick blood specimens, have reduced the fear and burden of testing on both test providers and test takers [23]. As HIV tests are now available in a wider variety of settings, they are more accessible to many [19]. In 2016, UNAIDS reported that an estimated 40% of PLWH did not know their status [7]. However, by 2020, this statistic had dropped to 16%.

In recent times, provider-initiated HIV testing and counseling (PITC) and family-based index HIV testing (FBIT) have supplanted VCT as an approach to HIV control and prevention. In 2007, the WHO and UNAIDS issued new guidelines and recommended countries and organizations to adopt PITC for HIV testing [24]. PITC involves health care providers offering routine HIV testing and counseling services to patients at a health facility irrespective of the purpose of their visit [24] with the aim of increasing HIV testing uptake and individual HIV status knowledge, so infected persons can be linked to medical care and support services [24,25]. Research from sub-Saharan Africa shows a high acceptability rate for PITC [26,27,28,29]. In 2004, Botswana was the first African country to introduce PITC in a widespread, systematic manner. Data from the first two and a half years of program implementation in the country revealed a great increase in testing: from 60,846 people tested in 2004 to 157,894 in 2005, and to 88,218 in 2006 [30]. The introduction of PITC in South Africa has also increased the proportion of new patients presenting at sexually transmitted infection clinics [28], boosted the number of children tested for HIV and put on ART in Malawi [31], and increased knowledge of partner HIV status in Uganda. 

Family-Based Index HIV Testing (FBIT) is a confidential and voluntary process that allows health care providers to ask index HIV-positive patients to list all their children and sexual partners. If the index patient agrees, contact is made with the persons listed and HIV testing and counseling services (HTS) are offered [32]. The FBIT approach has been identified as a significant global strategy that focuses on bridging testing gaps and increasing coverage of HTS for adults children, adolescents, and key populations who are usually missed when only the VCT approach is used [33]. The approach has also been found to increase the identification of HIV positive cases compared to the PITC approach [34].

### 2.2. Antiretroviral Therapy

Antiretroviral therapy is a pharmacological preparation designed to target HIV at key points of its life cycle [35]. Its aim is to minimize the risk of HIV and to reduce HIV viral load in blood, semen, and genitalia [36]. In 1987, the United States (US) Food and Drug Administration (FDA) approved the first ART drug, zidovudine, to treat HIV [37]. Zidovudine is a nucleoside reverse transcriptase inhibitor (NRTI), which upon phosphorylation, competes with free floating cytoplasmic nucleotides during the conversion of viral RNA into DNA [38]. This mechanism results in the termination of viral DNA formation. In 1995, the US FDA approved another ART, saquinavir, a protease inhibitor required for inhibiting the replication of HIV, and in 1996, nevirapine, a non-nucleoside reverse-transcriptase inhibitor (NNRTI) was approved by the US FDA [39]. Following the establishment of the President’s Emergency Plan for AIDS Relief (PEPFAR) in 2003 by George W. Bush, enfuvirtide, another ART, which is an entry and fusion inhibitor, was approved. Since its launch, PEPFAR has supported life-saving ART for over 20 million people and prevented millions from acquiring HIV in over fifty countries across the globe, especially those in sub-Saharan Africa [40]. In 2012, the first integrase inhibitor, raltegravir, followed by emtricitabine-tenofovir, which are both used as treatments for preexposure prophylaxis (PrEP), were also approved [41]. Due to ART, the life expectancy of the about 38 million people living with HIV in 2018 globally, was close to that of people not living with HIV [42].

Antiretroviral therapy can prevent HIV transmission. In 2010, the Pre-Exposure Prophylaxis Initiative (iPrEx) conducted a double-blinded, placebo-controlled Phase III clinical trial. Results from the trial showed that when taken every 24 h, PrEP can cause a 44% reduction in HIV infection risk and can also reduce infection rates by 73% [43]. In 2011, the HIV Prevention Trials Network (HPTN 052) also conducted a phase III two-arm randomized controlled multi-center trial to ascertain the potential of ART to prevent sexual transmission of HIV-1 in serodiscordant couples. Study findings reported in 2015, showed a 93% reduction in HIV transmission among 1171 serodiscordant couples [44]. Results from the most recent HPTN 084 study in 2020 showed the efficacy of the PrEP cabotegravir, an extended-release injectable suspension, when administered once every two months [45]. Consequently, on 20 December 2021, the US FDA licensed cabotegravir for at-risk adults and adolescents who weigh at least 35 kg [46]. Cabotegravir is administered as two initiation injections: a month apart and subsequently, every two months.

In July 2020, the antiretroviral (ARV) medication dapivirine was incorporated into the dapivirine vaginal ring (DVR). The ring is made of silicone and infused with ARV. It slowly releases ARV medication throughout the month and is known to reduce the risk of HIV infection by 61% [47]. In March 2022, the South African Health Products Regulatory Authority (SAHPRA) approved DVR for use by women aged 18 years and older to reduce their HIV risk [48]. It is important to note that when ART, whatever the form, is used consistently as prescribed by a physician, the HIV viral load in blood, sperm, vaginal fluid, and rectal fluids can be significantly reduced to undetectable levels. This is referred to as viral suppression or an ‘undetectable’ viral burden [49]. At this level, the risk of transmission is absent or reduced significantly.

#### 2.2.1. Trends in Antiretroviral Therapy Coverage across WHO Regions

There are six WHO regions: Africa, Americas, Eastern Mediterranean, Europe, South-East Asia, Western Pacific, and South-East Asia. According to the WHO, approximately 26 million PLWH are in the Africa Region and account for about 70% of all AIDS-related mortalities globally. About 3.7 million people live with HIV in the Americas, 3.7 million in South-East Asia, 2.6 million in Europe, 420,000 in the Eastern Mediterranean, and 1.9 in the Western Pacific [50]. In 2000, 600,000 PLWH were on ART compared to 2.0 million in 2005, and 7.8 million in 2010. It is projected that by 2030, an estimated 33.0 million PLWH will be on ART globally. The ART trend analysis discussed in this section is based on data extracted from UNAIDS data estimates from 2000 to 2019 [51].

##### Africa

Per available data, Botswana had the highest significant annual ART coverage increase of 4.8% from a baseline coverage of zero in 2000, to 82% at the end of 2019 (*t* = 40.9, *p* < 0.001) (Figure 1). The annual ART coverage in Ghana was the lowest in 2000 (2.3%) with an increase to 45% at the end of 2019 (*t* = 15.8, *p* < 0.001). Ethiopia and South Africa also experienced significant annual ART coverage increases of 4.4% and 4.2% respectively from a baseline coverage of zero in 2000, to 74% and 70% respectively in 2019 [51].

##### Americas

Peru had the highest significant annual ART coverage increase of 4.2% from a baseline coverage of zero in 2000, to 69% by the end of 2019 (*t* = 25.4, *p* < 0.001) (Figure 1). Chile’s ART coverage was at 10% in 2000 and rose significantly by 2.8% annually to 68% by the end of 2019 (*t* = 32.2, *p* < 0.001). Cuba, Haiti, and Brazil also saw significant increases in their annual ART coverage as well in 2019—3.6% (*t* = 30.3, *p* < 0.001), 3.9% (*t* = 16.9%, *p* < 0.001), and 2.9% (*t* = 8.9, *p* < 0.001) respectively [51].

##### Eastern Mediterranean

Antiretroviral therapy coverage for each of the five countries in the WHO Eastern Mediterranean region was less or equal to 25% (Figure 1). Overall, Iran had the highest annual ART coverage. Iran saw a steady progress of 1.3% every year from a baseline coverage of zero in 2000 to 25% by 2019 (*t* = 10.3, *p* < 0.001). The annual ART percentage increase for Pakistan and Afghanistan in 2019 was 0.4% (*t* = 5.4, *p* < 0.001) and 0.5% (*t* = 8.6, *p* < 0.001) respectively. Syria and Yemen also experienced significant annual increases in ART coverage of 1.1% (*t* = 10.4, *p* < 0.001) and 1.4% (*t* = 15.6, *p* < 0.001) from 2000 to 2019 [51].

##### Europe

In 2000, France had an ART coverage of 70%, which progressed steadily by 0.3% annually to 82% by the end of 2019 (*t* = 2.8, *p* = 0.011) (Figure 1). Also in 2000, Spain and the Netherlands had an ART coverage of 32% and 43% respectively which increased to 85% and 87% coverage by 2019. Armenia had the highest significant annual ART coverage increase in 2000. Beginning at zero percent coverage and progressing steadily at 5.1% annually to end up at 85% coverage in 2019 (*t* = 16.3, *p* < 0.001). Of the five countries in the WHO European region, Ukraine had the lowest annual ART coverage of 54% at the end of 2019. Ukraine’s coverage increased significantly by 2.8% every year from 2000 to 2019 (*t* = 10.5, *p* < 0.001) [51].

##### Western Pacific

In the WHO Western Pacific region, Cambodia had the highest significant annual ART coverage of 5.1% from zero percent in 2000 to 84% in 2019 (*t* = 27.4, *p* < 0.001) (Figure 1). Australia’s ART coverage was highest (47%) in 2000 and progressed steadily to 83% in 2019, accounting for an annual increase of 2.1%. The annual increase for New Zealand, Malaysia, and the Philippines were all significant beginning from zero percent in 2000 to 63%, 50%, and 44% respectively by the end of 2019 [51].

##### South-East Asia

Thailand’s ART coverage saw the highest significant annual increase of 4.3% from a coverage of zero percent to 80% by the end of 2019 (Figure 1). By the end of 2019, ART coverage in Myanmar and Nepal had increased significantly to 76% and 63%, respectively [51].

### 2.3. HIV-Related Stigma

HIV-related stigma is the prejudice, negative attitudes, and abuse directed at PLWH. It is a multidimensional construct and manifests in various forms including discrimination, shunning by family members, peers and the wider community, the erosion of rights, violence, and psychological damage. HIV-related stigma causes people to be labeled as socially unacceptable [52], and as a result, has negative implications for HIV testing uptake, treatment, and the utilization of HIV services. HIV-related stigma is rooted in the fear of HIV and misconceptions about how the disease is transmitted and what it means to live with HIV. People’s perceptions of HIV as being associated with a certain group of people, has caused negative judgments to be placed on all PLWH [52].

There are various types of HIV-related stigma. Self-stigma associated with HIV is a stigma that affects the mental wellbeing of PLWH or key populations. It breaks down their confidence, builds a wall of silence and shame, a sense of worthlessness, and limits meaningful self-agency, adherence to treatment, and the utilization of HIV services [53]. Governmental stigma associated with HIV occurs when a country’s laws and policies on HIV discriminate, alienate, and exclude PLWH. Health care stigma associated with HIV occurs in health care settings and includes mandatory HIV testing without patient consent or appropriate counseling, minimized health care provider contact with PLWH, delayed or denied treatment, and the isolation of PLWH from other patients. Employment stigma associated with HIV causes PLWH to suffer social isolation and ridicule from their co-workers and employers, or to experience discriminatory practices such as termination or refusal of employment. Community-level stigma and discrimination force PLWH to leave their homes and to change their daily activities. The isolation and social rejection can lead to low self-esteem, depression, and even thoughts or acts of suicide. The elimination of all forms of HIV-related stigma is important in the fight against HIV as it impacts HIV prevention efforts.

Unlike VCT where the utilization of HIV testing and counseling services are based on a person’s volition, PITC gives people the option to either opt-in or opt-out of testing. This approach has reduced the stigma associated with HIV testing and thus, encouraged people to get tested without the fear of being stigmatized. In the past when a person was offered an HIV test, the perception was that a health care provider had noticed a symptom attributable to HIV. With PITC, things are changing; people now feel more at ease to openly talk about HIV, and to ask questions about care even in public settings.

## 3. Global HIV Prevention Challenges

### 3.1. Voluntary Counselling and Testing

Regardless of the expansion of VCT services globally, challenges persist [7,10,18]. Due to financial constraints, the cost of investment, and the lack of medical infrastructure, laboratory and trained health care providers, VCT is not a national priority for many governments, especially those in sub-Saharan Africa. As most of these countries depend on donor support to fund VCT services, they find it difficult to make VCT accessible to their entire population [10,11].

Structural barriers such as the distance between one’s residence and the nearest public health facility as well as delays in the reception of HIV test results due to the high workload and burnout of health care providers, have been linked to declining VCT uptake [54]. Indeed, increased complaints of health care provider burnout in Kenya, Uganda, Zambia, and Zimbabwe almost disrupted VCT services [55]. In countries where lay counselors are used to provide HIV counseling services to reduce the workload of health care providers, precarious working conditions and the financial instability of the lay counselors has made the strategy unsustainable [54].

Low levels of awareness of the benefits of VCT services in developing countries is another challenge that impedes VCT effectiveness. In 2014, UNAIDS reported that 19 developing countries were experiencing challenges in raising community awareness of the benefits of VCT [56]. That year, Nigeria and Kenya had an awareness level of 34% and 41% respectively, compared to an average level of 72% in developed countries [57]. The fear of stigma, lack of test confidentiality, and poor treatment from health care providers serve as further challenges to global VCT testing uptake [18,58]. The stigmatization and criminalization of HIV has caused certain populations (lesbians, gay, bisexuals, transgender or intersex, drug users, men who have sex with men (MSM), sex workers, and sexually active young people) to refrain from getting tested for fear of being diagnosed with HIV, which could result in maltreatment, extortion, discrimination, violence, or arrest [56]. In a study conducted on 45,000 gay men in 28 sub-Saharan African countries including Nigeria, Malawi, and Kenya, researchers found that only 50% of study participants had tested for HIV in the past year. The researchers attributed the low VCT uptake to the anti- lesbian, gay, bisexual, transgender, and queer (LGBTQ) laws, which they said promote discrimination and stigmatization of HIV among this population [55]. The results of a study conducted in sub-Saharan African on factors that prevent HIV testing showed that the cost of HIV testing negatively impacts VCT access and uptake [9,10,21]. The coronavirus disease of 2019 (COVID-19) has also negatively impacted VCT uptake. Owing to a national lockdown as a measure to contain the spread of COVID-19, VCT services and ART initiation were interrupted in 65 clinics in South Africa [59].

Regardless of its benefits, the comprehensive implementation of PITC in sub-Saharan Africa is woefully inadequate [60] due to inadequate training and shortage of trained health care workers, increased workload of health care providers, and inadequate space for providing confidential counseling services [61,62]. A study conducted to assess institutional capacity to implement PITC in Zimbabwe found that while most sites had staff, they were not many, and most of the testing was done by nurses, with laboratory technologists providing quality control. In situations where laboratory technologists conducted the test, patients experienced delays in receiving their results, causing some not to return to facilities for their results [63]. Regarding space, only 10 of the 16 sites involved in the study had adequate space to provide confidential counseling services [63]. Currently, there are global debates on patient perceptions of PITC. Although several studies have documented high patient acceptance of the approach, it is unclear the extent to which this is accurate, as intentional and unintentional coercion by health care providers, as well as other factors, may influence patient decision-making at the point of testing [30].

Despite the success of the FBIT approach, there are some barriers that impact its extensive utilization. These barriers include the difficulty of people disclosing their HIV status to family members, and the fear of psychological instability especially among children [64,65,66,67]. In a study conducted in Ghana, it was found that despite participants’ willingness to provide information about family members, they were fearful of possible stigmatization and discrimination once their HIV status was disclosed [68]. Fear was found to be a barrier to the implementation of the FBIT approach [69].

### 3.2. Antiretroviral Therapy

Overall, global estimates show that regardless of the significant progress made in ART coverage, the lack of adequately trained health care providers who can effectively dispense and monitor ART in some sub-Saharan African countries serve as a major drawback in the scaling up of ART coverage [70]. In some sub-Saharan African countries like Ghana, the shortage of ART, and the non-adherence to ART by PLWH are the major challenges [71].

In the WHO Eastern Mediterranean region, a “testing gap” (the gap between the estimated number of PLWH and the number of PLWH who know that they are infected), as well as the region’s vast reliance on donor support, have been identified as responsible for low ART coverage. In Latin America and the Caribbean regions, the lack of medication adherence associated with substance abuse, HIV- related stigma, depressive symptoms, and high pill burden are the challenges [72]. While the WHO European region has one of the lowest HIV prevalence rates, it has the highest ART coverage. However, unlike some PLWH in low and -middle-income countries who have free access to ART, PLWH in France have to pay for ART, which can be expensive [73].

In the WHO Western Pacific region, the Philippines has the highest HIV transmission rate and yet, the lowest ART coverage. This is due to the uneven distribution of HIV treatment centers across the country’s approximately 7000 islands [74]. Despite the launching of a 2025 agenda to end HIV transmission, PLWH in Australia are challenged with the high cost of ART and HIV-related stigma, and in the WHO South-East Asia region as in the Eastern Mediterranean region, the over dependence on donor assistance, which has been dwindling over the years [75], has negatively impacted ART coverage.

As of June 2018, 46 countries had obtained regulatory approval for PrEP, and 39 of those countries have included PrEP usage within their HIV policies [76]. Figure 2 shows trends in global PrEP regulatory approval. In the Americas, the United States, Canada, and Brazil have obtained regulatory approval for PrEP. In the European region, PrEP is available in only Norway, France, Portugal, Belgium, and the United Kingdom. In the Western Pacific region only Australia has obtained regulatory approval for PrEP utilization. In the South-East Asia region, two countries, Thailand and Malaysia have obtained regulatory approval for PrEP utilization. Out of the 54 countries in Africa, only four (South Africa, Namibia, Kenya, and Zimbabwe) have obtained regulatory approval for PrEP [77].

Although PrEP has been approved in the aforementioned countries, availability is sometimes restricted to those who need it due to a number of factors including the lack of clinical guidelines or policies for the administration of the drug, which makes doctors reluctant to prescribe, as well as the non-inclusion of PrEP as a free medication in national health plans. This situation causes some people to turn to unregulated online sellers for the prophylaxis [77]. Governments in all regions of the world need to expedite PrEP regulatory processes, so the medication can be available to all who need it. They also need to subsidize the cost of PrEP, so it is affordable [76].

### 3.3. HIV-Related Stigma

Perceptions of HIV as a disease associated with certain key populations (MSM, people who inject drugs, and sex workers), the notion that the disease is solely transmitted through sex and is the result of personal irresponsibility that deserves to be punished, have contributed to the stigma and discrimination associated with the disease. The stigma and discrimination key populations experience, drive them to the margins of society where the fear of rejection or even violence, makes getting tested, disclosing their HIV status, or accessing HIV treatment very difficult [78]. According to a study conducted in 2016, 60% of MSM and people who inject drugs in Europe reported having experienced discriminatory attitudes from health care professionals, which negatively impacted the quality of HIV treatment and care they received [79]. In their report on HIV-related stigma and its effects on HIV prevention, UNAIDS and the WHO underscored the fact that the fear of stigma and discrimination are the main reasons why people are reluctant to get tested [80]. These challenges they observed, contribute to late diagnosis, the onset of AIDS, and an increase in the global HIV burden [80].

HIV-related stigma prevents many young women from using HIV medications that could otherwise protect them from getting HIV. In an HIV study conducted in South Africa, many young women reported being afraid to use vaginal gels and pills for the fear of being mistakenly identified by others as having HIV [81].

In 2014, about 72 countries including the US had some form of legislation in place to prosecute PLWH for a range of offenses including the non-disclosure of their HIV status to sex and needle sharing partners [82]. In countries like China where sex work is partially legal, the law rarely protects sex workers, putting them at risk for discrimination, abuse, and violence from both state and non-state actors such as law enforcement, partners, family members, and their clients [83]. The criminalization of HIV undermines public health efforts to prevent and reduce HIV transmission, in that it punishes PLWH and causes people at risk for HIV to feel stigmatized and unwilling to utilize services that support HIV prevention, diagnosis, and treatment [84]. Research has also shown that the criminalization of HIV does not necessarily prevent its transmission [85].

## 4. The Way Forward

### 4.1. Funding and Staffing

If VCT uptake and ART coverage are to be expanded globally, then it is imperative for national governments of countries in deficit to prioritize them as part of their HIV prevention strategies and to set aside enough funds from their national budgets to meet the needs of their citizens and PLWH. The reliance on development partners for aid to fund such services is not sustainable, as aid will only be provided when development partners have excess funds to spare.

Scaling up VCT means addressing the critical shortage of trained VCT laboratory technicians and counselors, and providing training, support, and supervision of health care personnel to avoid staff burnout and increased workload, and ensuring accuracy and prompt communication of test results [22]. It also requires locating VCT services and HIV clinics closer to communities (both rural and urban) so they are more accessible, and training and remunerating volunteer HIV counselors, so they can support health care providers in the delivery of counseling services. With regards to ART, appropriate health care providers need to be recruited and trained in how to correctly dispense ART, so there is no wastage. National governments of countries that experience ART shortage need to invest in, acquire and store enough ART so they are available at all times to PLWH.

The WHO recommends the rational reallocation of tasks between different categories of healthcare workers ranging from basic training to higher training. This is known as task sharing [11]. Task sharing has been shown to increase the efficiency of healthcare workers and health outcomes in communities [12]. Although tasking sharing has been implemented in the health sector for many years, its scope has not encompassed HIV counseling and testing (HCT) [5,11].

### 4.2. Special Populations

To increase VCT uptake, issues related to stigma, discrimination, and marginalization in general and among key populations in particular need to be addressed. This can be achieved through a simple rapid assessment to collect data that can be used to develop an intervention to meet the express VCT needs of this population [58] and to improve upon the relationship between PLWH and health care providers.

### 4.3. Awareness Creation and Promotion of Benefits

So as to increase population willingness to participate in HIV VCT, national governments need to share information about the positive effects HIV testing is having on the reduction of the disease in their respective countries. Public awareness mechanisms such as billboards, radio programs, public service announcements, and bus advertisements could be used to share the benefits. PLWH need to be educated on the benefits of taking ART, adhering to regimens, and the complications that could arise from nonadherence.

### 4.4. HIV-Related Stigma

At the individual level, PLWH need to be educated and provided with information on how to protect themselves and their human rights. At the community level, social and structural interventions to change societal beliefs about PLWHA need to be implemented. At the national level, governments need to create safe spaces for people to talk publicly about HIV as doing this will help to normalize the subject and correct any misconceptions about the disease. Countries also need to consider repealing, modernizing, or deprioritizing obsolete HIV criminalization laws. At the health systems level, health care provider phobias about HIV must be addressed, as the fear of acquiring the disease, causes some to take unnecessary and often stigmatizing actions.

## 5. Conclusions

HIV-related stigma has been found to negatively impact VCT uptake and ART coverage. Thus, unraveling the specific aspects of HIV-related stigma that impede VCT and ART use, and investing in HIV stigma reduction interventions could help to address some of the challenges associated with HIV prevention.

## Figures and Tables

**Figure 1 ijerph-19-06597-f001:**
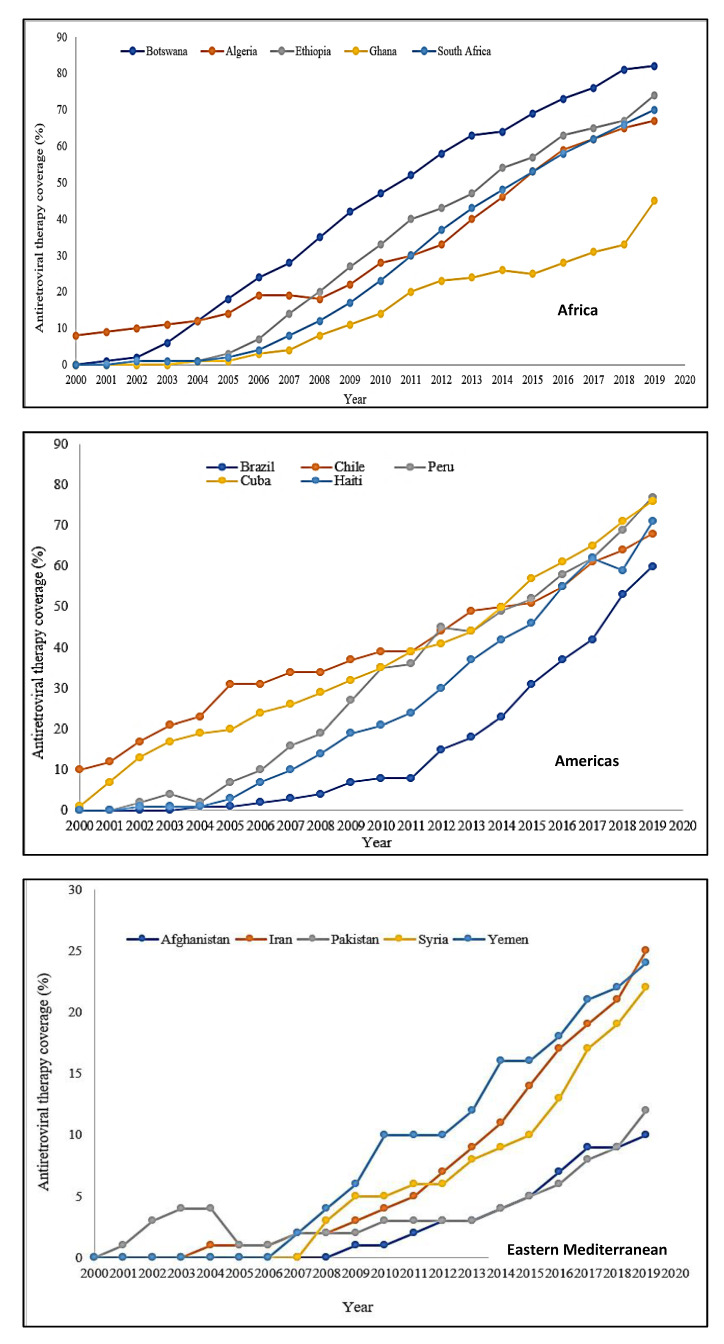
ART trends in the WHO regions based on data extracted from UNAIDS estimates for 2000 to 2019. Joint United Nations Programme on HIV/AIDS (UNAIDS) Estimate the World Bank Data. Available online: https://data.worldbank.org/indicator/SH.HIV.ARTC.ZS?end=2020&start=2000&view=char, accessed on 20 September 2021 [51].

**Figure 2 ijerph-19-06597-f002:**
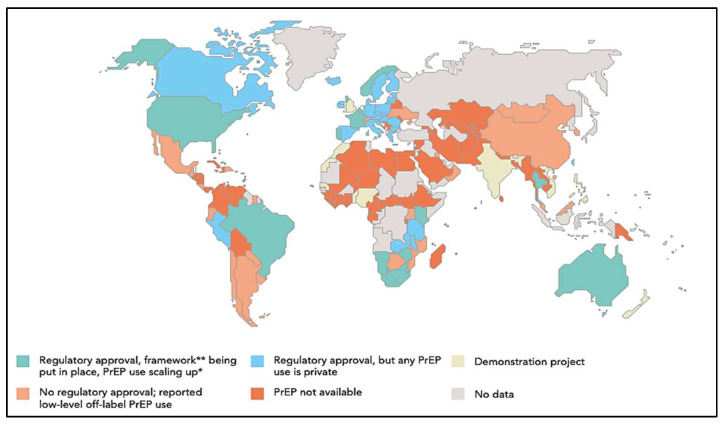
Availability of Pre-Exposure Prophylaxis by Country culled from United Nations Programme on HIV/AIDS. Ending AIDS—Progress Towards the 90-90-90 Targets. Available online: https://www.unaids.org/sites/default/files/media_asset/Global_AIDS_update_2017_en.pdf, accessed on 24 December 2021 [77]. * PrEP scale up among men who sleep with men; ** A framework for PrEP scale-up includes clinical guidelines; service provider training; access-oriented PrEP services; use of generic PrEP, price subsidy or reimbursement; effective demand creation.

## Data Availability

Not applicable.

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
