# Peer review of "Voluntary Counseling and Testing, Antiretroviral Therapy Access, and HIV-Related Stigma: Global Progress and Challenges"

_ijerph, 2022, doi:10.3390/ijerph19116597_

Round 1
Reviewer 1 Report
ijerph-1513807-peer-review
Thank you for the opportunity to review “Voluntary Counseling and Testing, Antiretroviral Therapy Access, and HIV-related Stigma: Global Progress and Challenges”. While a review of this sort has its merits, I do not feel the paper is currently suitable for publication and will need significant editing and additional scholarship.
Abstract: You don’t get to the topic until the last sentence. I would throw out this abstract and start fresh; we don’t need to know the history of HIV (which many of us can recite by heart) but do tell us the high level summary of progress and challenges of VCT, ART, and stigma
Introduction, first sentence: of course this is the case. In 1981 the disease was new with few cases – do you mean there are more people living with HIV now than when incidence was at its high water mark, in the 1990s and 00s?
Line 43-4: “While coverage in the developed world has significantly improved, the same cannot be said of sub-Saharan Africa where coverage is significantly lower.” ART coverage in SSA has certainly improved by leaps and bounds – even if gaps remain. I would reword this sentence.
Line 60+: “Originally implemented as an individual-level clinic-based procedure, VCT is now also implemented as a community-based approach with the aim of increasing uptake[10, 11].” Community and couples-based approach. Getting both members of a couple involved has been very important towards helping improve uptake and reducing stigma, and merits mention in this review. There are many examples of this, here are just a few to get you started:
- Kelley AL, Karita E, Sullivan PS, Katangulia F, Chomba E, Carael M, et al. Knowledge and perceptions of couples' voluntary counseling and testing in urban Rwanda and Zambia: a cross-sectional household survey. PLoS One. 2011;6(5):e19573.
- Allen S, Karita E, Chomba E, Roth DL, Telfair J, Zulu I, et al. Promotion of couples' voluntary counselling and testing for HIV through influential networks in two African capital cities. BMC public health. 2007;7:349.
- Wall KM, Inambao M, Kilembe W, Karita E, Chomba E, Vwalika B, et al. Cost-effectiveness of couples' voluntary HIV counselling and testing in six African countries: a modelling study guided by an HIV prevention cascade framework. J Int AIDS Soc. 2020;23 Suppl 3:e25522.
- Wall KM, Inambao M, Kilembe W, Karita E, Vwalika B, Mulenga J, et al. HIV testing and counselling couples together for affordable HIV prevention in Africa. Int J Epidemiol. 2019;48(1):217-27.
- https://www.cdc.gov/hiv/effective-interventions/diagnose/testing-together?Sort=Title%3A%3Aasc
Line 70: typo ? VCT *has* been implemented
Line 93: End your sentence after “Dropped to 16%.” As the remainder of the sentence is redundant.
Line 96: you make HIV sound like malaria. Recommend dropping the word “multistage”
Progress regarding ART. A big omission from this paper are the amazing strides we’ve made to show how ART can prevent onward transmission of HIV. Just a few examples include the Results of the Phase III iPrEx study that show that PrEP taken daily cuts infection rates by 73%. That same year, the results of HPTN052 shows that ART taken by the positive partner in discordant couple relationships dramatically reduces onward HIV transmission, by 96%. These two studies (and many other PrEP studies) drives international policy to provide ART to everyone with HIV, and to provide PrEP to groups at higher risk of HIV acquisition. New developments in PrEP include, in 2020, the results from HPTN 084 showing that a long-acting form of HIV prevention injected once every eight weeks prevented HIV acquisition in women when compared to the standard-of-care daily oral HIV PrEP.
2.2. on ART: Listing when drugs became available isn’t very interesting. It’s important to describe the global struggle to get ART to the people who needed it most, negotiating with drug companies to get competitive pricing, and supporting the health care systems to provide quality care for the volume of patients (particularly in SSA) that were going to need care. How and when did access begin to improve in SSA (and globally)? What were the hurdles faced, how were they met and resolved? You do this a little in 2.2.1 on trends in coverage – that is much more interesting. But I feel you can do a better job in describing trends rather than just focusing on the worst and best case scenarios in each WHO region. For example, what year did AIDS deaths actually begin to decline (due to ART)? How do trends of HIV incidence vary over time (and what impact might ART have had on this)? You also need to highlight the breakthrough c1996 of multidrug ART (no mention of it?) as this was a critical achievement to saving people’s lives, and an international travesty that it took ~10-20 years to get ART programs up and running in regions that needed them most. Some of these may fit better in section 3, where you describe challenges.
Lines 105-7: this sentence doesn’t make sense, as the two things don’t have anything to do with each other. PEPFAR was a MAJOR development in getting ART to the people who need them. Enfuvirtide is just a drug. You should spend more time than just a half-sentence describing PEPFAR! How did this affect access?
Figure 1: I would add a regional trend to these graphs. Showing only a few (why are they shown? What do these few countries represent?) countries isn’t helpful or informative
Line 196: “underperforming economy” is a rather stigmatizing phrase, and doesn’t capture the complexity of LMIC hardships. I recommend rephrasing (or just dropping that phrase).
A major point not mentioned at all in the paper is the disruption COVID-19 has brought to health care systems and HIV care globally. Service interruptions, regional and national shutdowns, fear of exposure to SARS CoV2 has negatively impacted the entire global economy, and not mentioning this in your review is a large omission. There’s been some debate on this, and a growing body of literature explores this, for example:
- The impact of the COVID-19 lockdown on HIV care in 65 South African primary care clinics: an interrupted time series analysis Dorward, Jienchi et al. The Lancet HIV, Volume 8, Issue 3, e158 - e165
- Essien EJ, Mgbere O, Iloanusi S, Abughosh SM. COVID-19 Infection among People with HIV/AIDS in Africa: Knowledge Gaps, Public Health Preparedness and Research Priorities. Int J MCH AIDS. 2021;10(1):113-118. doi:10.21106/ijma.461
- https://www.unaids.org/en/resources/presscentre/featurestories/2020/october/20201016_covid-impact-on-hiv-treatment-less-severe-than-feared
Author Response
Abstract: You don’t get to the topic until the last sentence. I would throw out this abstract and start fresh; we don’t need to know the history of HIV (which many of us can recite by heart) but do tell us the high level summary of progress and challenges of VCT, ART, and stigma. Done, see abstract
Introduction, first sentence: of course this is the case. In 1981 the disease was new with few cases – do you mean there are more people living with HIV now than when incidence was at its high water mark, in the 1990s and 00s? Done- see lines 29-30
Line 43-4: “While coverage in the developed world has significantly improved, the same cannot be said of sub-Saharan Africa where coverage is significantly lower.” ART coverage in SSA has certainly improved by leaps and bounds – even if gaps remain. I would reword this sentence. Done- see lines 45-46
Line 60+: “Originally implemented as an individual-level clinic-based procedure, VCT is now also implemented as a community-based approach with the aim of increasing uptake[10, 11].” Community and couples-based approach. Getting both members of a couple involved has been very important towards helping improve uptake and reducing stigma, and merits mention in this review. There are many examples of this, here are just a few to get you started: Done – see lines 63 and 64 and lines 67-70
Kelley AL, Karita E, Sullivan PS, Katangulia F, Chomba E, Carael M, et al. Knowledge and perceptions of couples' voluntary counseling and testing in urban Rwanda and Zambia: a cross-sectional household survey. PLoS One. 2011;6(5):e19573.
Allen S, Karita E, Chomba E, Roth DL, Telfair J, Zulu I, et al. Promotion of couples' voluntary counselling and testing for HIV through influential networks in two African capital cities. BMC public health. 2007;7:349.
Wall KM, Inambao M, Kilembe W, Karita E, Chomba E, Vwalika B, et al. Cost-effectiveness of couples' voluntary HIV counselling and testing in six African countries: a modelling study guided by an HIV prevention cascade framework. J Int AIDS Soc. 2020;23 Suppl 3:e25522.
Wall KM, Inambao M, Kilembe W, Karita E, Vwalika B, Mulenga J, et al. HIV testing and counselling couples together for affordable HIV prevention in Africa. Int J Epidemiol. 2019;48(1):217-27.
https://www.cdc.gov/hiv/effective-interventions/diagnose/testing-together?Sort=Title%3A%3Aasc
Line 70: typo ? VCT *has* been implemented Done- see line 76
Line 93: End your sentence after “Dropped to 16%.” As the remainder of the sentence is redundant Done- see line 99
Line 96: you make HIV sound like malaria. Recommend dropping the word “multistage” Done- see line 102
Progress regarding ART. A big omission from this paper are the amazing strides we’ve made to show how ART can prevent onward transmission of HIV. Just a few examples include the Results of the Phase III iPrEx study that show that PrEP taken daily cuts infection rates by 73%. That same year, the results of HPTN052 shows that ART taken by the positive partner in discordant couple relationships dramatically reduces onward HIV transmission, by 96%. These two studies (and many other PrEP studies) drives international policy to provide ART to everyone with HIV, and to provide PrEP to groups at higher risk of HIV acquisition. New developments in PrEP include, in 2020, the results from HPTN 084 showing that a long-acting form of HIV prevention injected once every eight weeks prevented HIV acquisition in women when compared to the standard-of-care daily oral HIV PrEP. Done- see lines 119-129
2.2. on ART: Listing when drugs became available isn’t very interesting. It’s important to describe the global struggle to get ART to the people who needed it most, negotiating with drug companies to get competitive pricing, and supporting the health care systems to provide quality care for the volume of patients (particularly in SSA) that were going to need care. How and when did access begin to improve in SSA (and globally)? What were the hurdles faced, how were they met and resolved? You do this a little in 2.2.1 on trends in coverage – that is much more interesting. But I feel you can do a better job in describing trends rather than just focusing on the worst and best case scenarios in each WHO region. For example, what year did AIDS deaths actually begin to decline (due to ART)? How do trends of HIV incidence vary over time (and what impact might ART have had on this)? You also need to highlight the breakthrough c1996 of multidrug ART (no mention of it?) as this was a critical achievement to saving people’s lives, and an international travesty that it took ~10-20 years to get ART programs up and running in regions that needed them most. Some of these may fit better in section 3, where you describe challenges. Done- see lines 275-292
Lines 105-7: this sentence doesn’t make sense, as the two things don’t have anything to do with each other. PEPFAR was a MAJOR development in getting ART to the people who need them. Enfuvirtide is just a drug. You should spend more time than just a half-sentence describing PEPFAR! How did this affect access? Done- see lines 113-115
Figure 1: I would add a regional trend to these graphs. Showing only a few (why are they shown? What do these few countries represent?) countries isn’t helpful or informative Done- see page 8 – figure 2
Line 196: “underperforming economy” is a rather stigmatizing phrase, and doesn’t capture the complexity of LMIC hardships. I recommend rephrasing (or just dropping that phrase). Done- see line 215. Deleted the phrase
A major point not mentioned at all in the paper is the disruption COVID-19 has brought to health care systems and HIV care globally. Service interruptions, regional and national shutdowns, fear of exposure to SARS CoV2 has negatively impacted the entire global economy, and not mentioning this in your review is a large omission. There’s been some debate on this, and a growing body of literature explores this, for example: Done- see lines 248-251
The impact of the COVID-19 lockdown on HIV care in 65 South African primary care clinics: an interrupted time series analysis Dorward, Jienchi et al. The Lancet HIV, Volume 8, Issue 3, e158 - e165
Essien EJ, Mgbere O, Iloanusi S, Abughosh SM. COVID-19 Infection among People with HIV/AIDS in Africa: Knowledge Gaps, Public Health Preparedness and Research Priorities. Int J MCH AIDS. 2021;10(1):113-118. doi:10.21106/ijma.461
https://www.unaids.org/en/resources/presscentre/featurestories/2020/october/20201016_covid-impact-on-hiv-treatment-less-severe-than-feared
Reviewer 2 Report
The manuscript is well-written and aims to discuss global successes and challenges in HIV prevention efforts in the past four decades, focusing on VCT, ART, and HIV-related stigma. The introduction, the global HIV prevention efforts, and progress made after decades are good. The ART Coverage Across WHO Regions section based on data extracted from UNAIDS data analysis estimates from 2000 to 2019 is good. It gives an overview of the trends around the world. However, a few graphics should be revised to include legends (Africa and Europe, for instance); curves are not linked to a specific country.
Section 3, bringing the HIV prevention Challenges, focusing on VCT, ART, and Stigma are updated and well discussed. However, I missed information about an important prevention tool like Pre-exposure prophylaxis (or PrEP). I think the authors should have written at least a few paragraphs to describe the current PrEP situation worldwide as an essential preventive tool against HIV infections. It is imperative to encourage those at risk for HIV to talk about VCT, condom usage, PrEP, and ART so that they can take action to protect themselves and their partners.
Regarding the English language, minor spell checks are required. Please find below some of them: in blue, minor suggestions and yellow corrections to be made.
Line 17_ have been significantly (more) impacted than…
Line 88_ in “resource-poor settings” (p.116)[16].
Line 97_ Its aim is to minimize the risk of HIV (infection) and to reduce (the HIV) viral load of the infection in the blood, semen, and the genitalia[ 19]…
Line 207_ (inorder) to allow them to focus on other aspects of clinical care. However, the precarious 207working..
Line 236_ In the WHO Eastern Mediterranean region, a “testing gap” (the gap between….
Line 311- PLWH (heed) to be educated on the benefits of taking ART…
Line 318_correct any (mosconceptions)…
Line 320_ obsolete HIV (criinaization) laws. At the health systems level, health care provider….
Author Response
The ART Coverage Across WHO Regions section based on data extracted from UNAIDS data analysis estimates from 2000 to 2019 is good. It gives an overview of the trends around the world. However, a few graphics should be revised to include legends (Africa and Europe, for instance); curves are not linked to a specific country. Done- see page 5
Section 3, bringing the HIV prevention Challenges, focusing on VCT, ART, and Stigma are updated and well discussed. However, I missed information about an important prevention tool like Pre-exposure prophylaxis (or PrEP). I think the authors should have written at least a few paragraphs to describe the current PrEP situation worldwide as an essential preventive tool against HIV infections. It is imperative to encourage those at risk for HIV to talk about VCT, condom usage, PrEP, and ART so that they can take action to protect themselves and their partners. Done, see lines 119-129
Regarding the English language, minor spell checks are required. Please find below some of them: in blue, minor suggestions and yellow corrections to be made. Line 17_ have been significantly (more) impacted than… Done, see lines 13-14
Line 88_ in “resource-poor settings” (p.116)[16]. Done, see line 96
Line 97_ Its aim is to minimize the risk of HIV (infection) and to reduce (the HIV) viral load of the infection in the blood, semen, and the genitalia[ 19]… Done, see lines 102
Line 207_ (inorder) to allow them to focus on other aspects of clinical care. However, the precarious 207working.. Done, see line 226
Line 236_ In the WHO Eastern Mediterranean region, a “testing gap” (the gap between….
Done, see line 259
Line 311- PLWH (heed) to be educated on the benefits of taking ART… Done, see line 357
Line 318_correct any (mosconceptions)… Done, see line 364
Line 320_ obsolete HIV (criinaization) laws. At the health systems level, health care provider… Done, see line 366
Reviewer 3 Report
Dear Authors,
I have had the opportunity to read your manuscript with great pleasure.
I think the paper focuses properly on the current issues of HIV pandemic, as implementing testing and testing, optimizing the access to antiretroviral treatments, reducing the feeling of stigma which hevily impacts on special population.
In regarding to the last point, we have recently performed a survey among Italian PLWHIV that has highlighted how this population still feels strongly stigamtized. For this special patients, the infectious disease specialist is the only figure that manages all HIV and non-HIV related issue. Implementing the relationship between PLWHIV and General Practitioners could be another point that could be added to your discussion.
A minor observation: please add the extensive spelling of the abbreviagtions contained in the abstract (i.e. VCT line 20)
Author Response
In regarding to the last point, we have recently performed a survey among Italian PLWHIV that has highlighted how this population still feels strongly stigamtized. For this special patients, the infectious disease specialist is the only figure that manages all HIV and non-HIV related issue. Implementing the relationship between PLWHIV and General Practitioners could be another point that could be added to your discussion. Done, see line 352
A minor observation: please add the extensive spelling of the abbreviagtions contained in the abstract (i.e. VCT line 20) Done, see line 16